# Breaking the Silence: Barriers to Error Disclosure Among Nurses in the Kingdom of Saudi Arabia—A Cross-Sectional Study

**DOI:** 10.3390/healthcare13070831

**Published:** 2025-04-05

**Authors:** Naglaa Abdelaziz Mahmoud Elseesy, Budoor Ahmad Almezraq, Duaa Amr Hafez, Ohood Felemban, Ebaa Marwan Felemban

**Affiliations:** 1Public Health Nursing Department, Faculty of Nursing, King Abdulaziz University, Jeddah 21589, Saudi Arabia; naalsesei@kau.edu.sa (N.A.M.E.); dahafez@kau.edu.sa (D.A.H.); ofelemban@kau.edu.sa (O.F.); 2Nursing Administration Department, Faculty of Nursing, Alexandria University, Alexandria 21527, Egypt; 3Nursing Quality, Nursing Administration, Akhfji General Hospital, Dammam 32253, Saudi Arabia; moons.ezt@hotmail.com

**Keywords:** barriers, error disclosure, nurse error disclosure, Saudi Arabia, patient safety incident, adverse event, medical error report, medical error

## Abstract

Background: Medical errors are common, and in particular, medication errors are one of the leading causes of morbidity and mortality in healthcare. Nurses must disclose errors to build trust and ensure patient safety despite communication barriers and fear of reprisals. Truthful documentation and better collaboration can improve patient outcomes. Aim: The aim of this study was to assess barriers to error disclosure among nurses in the Kingdom of Saudi Arabia. Methods: A descriptive cross-sectional survey was conducted in the months of June and August 2024 using a convenience sample of 255 nurses at King Fahad Hospital (KFH), Hofuf. A self-administered questionnaire that contained socio-demographic questions, as well as the Barriers to Error Disclosure Assessment (BEDA) tool, was deployed. Results: The majority of the sample were female (92.2%) and Bachelor’s degree holders (80.8%) who had 1–5 or 6–10 years of experience, representing 45.5% and 29.0% of the sample, respectively. When medical errors occur, only 18% of the nurses disclosed them to patients and their families, and 34.5% of the respondents disclosed medical errors to another team of healthcare professionals. The types of barriers to error disclosure were identified as relating to confidence and knowledge barriers, institutional barriers, psychological barriers, and financial concern barriers. The overall barrier score had an average of 63%. The barriers with the highest scores were those relating to psychological barriers (68.2%), followed by institutional barriers (66.5%) and financial concern barriers (64.5%). Conclusions: Targeted interventions are required for addressing the identified specific needs to support healthcare providers, specifically nurses. New regulations and policy changes are crucial for training programs implementation, enhancing safety culture, and tackling job-related insecurities to minimize barriers to error disclosure and ultimately provide better patient care quality. Further investigations may include a different approach, and it is recommended to provide deeper insights into nurse experiences.

## 1. Introduction

Disclosing errors in healthcare is vital for the safety of patients. When an error occurs, healthcare providers, including attending nurses, should be honest and offer patients and their families complete and detailed information about the incident. This honesty serves as a dual purpose of assisting in settling any further damage from occurring and contributing to trust building in the nursing profession [1]. To establish a successful and thorough system for error disclosure in healthcare, it is crucial to systematically pinpoint and address the barriers that prevent healthcare providers (HCPs) from disclosing errors effectively [2].

A number of barriers hinder this approach to transparent disclosure from being implemented, including ineffective training and a lack of communication skills. While other supporting reasons have been explored, miscommunication between HCPs and the patients they treat is still believed to be the primary cause prompting lawsuits against them [3]. Furthermore, research-based evidence indicates that there are various barriers that significantly impede HCPs’ preparedness to disclose errors, including fear of legal consequences, professional reputation concerns, cultural and organizational factors, emotional and psychological barriers, fear of professional repercussions, lack of clear policies [2], and the absence of training, guidance, or experience in communication related to disclosure [4].

There are also institutional barriers that impede the disclosure of medical errors; these include unsupportive institutional cultures and a lack of clarity about policies concerning the disclosure of errors [5]. In their study, Ragucci K et al. (2016) showed the importance of involving qualified and trained professionals in the error disclosure policy of a health organization [6]. By providing guidance and assistance while error disclosures are performed, patient welfare and effective error handling can be ensured [7].

Moreover, the decision to disclose certain information may be influenced by a number of psychological processes. These include fear of insult, feelings of guilt and embarrassment, and the belief that the negative perceptions of colleagues, in addition to patient perceptions, will compromise them in terms of respect and trust [7]. Another fear-driven factor is the fear of being sued, concerns over losing their malpractice coverage, and considerations of having to pay increased premiums for malpractice insurance [8]. Birks Y et al. (2015) concluded that these financial consequences can create barriers to the open disclosure of information amongst HCPs [8].

In fact, a regular supply of education and training among health workers is one of the strategies that encourages the successful disclosure of medical errors. Improving medical error disclosure may include offering disclosure training to the healthcare team and creating a transparent environment. Studies indicate that specific training on the disclosure of medical errors enhances attitudes, skills, and knowledge concerning the same in medical professionals [9].

For this, collaboration between HCPs will be the key to improving patient outcomes and experiences. However, team-based care can also become a source of encouraging the disclosure of medical errors among team members of healthcare workers (HCWs). Therefore, it is indeed a good approach for every HCP, even if they are working in a supportive role, to contribute toward enhancing medical error disclosure. According to a study by Yan S et al. (2017), actively engaging every level of healthcare in disclosure initiatives is critically important [10]. Additionally, there are conflicting opinions regarding the timing of error disclosures: Some believe one should only disclose when there is severe patient harm, while others advocate disclosure for near-miss incidents [7]. Nurses hold a certain anxiety that being transparent about errors may give the idea that they were unsafe during patient care and could potentially ruin patient trust in healthcare providers [1].

The central aim of this research inquiry is to identify the barriers to error disclosure for the purpose of building safer healthcare environments by encouraging open and honest communication in error disclosure, thus creating the conditions for the ethical reporting of errors.

### 1.1. Research Problem

The Disclosure of Patient Safety Incidents (DPSIs) is a systematic and practical approach that addresses communication strategies following a patient safety incident. The DPSI outlines the following procedures: First, in the event of a patient safety incident, the attending medical professional will promptly inform the patient and caregiver about the situation, express sympathy and regret, and commit to investigating the cause of the incident. Second, a formal apology will be offered if a medical error is found to be the cause of the incident. Third, compensation will be provided in line with the extent of the damage caused by the incident. Final step, a pledge is made to prevent similar incidents from occurring in the future. As a result, in order to advance the DPSI, a number of nations and organizations have incorporated patient safety incident communication principles into their accreditation requirements [1]. Moreover, there are policies that promote heightened transparency, and a more profound understanding of the factors that support successful open disclosure is essential for practical implementation [11].

Many studies have observed that nurses are seldom involved in the DPSI. This might be attributed to the fact that nurses usually deal with problems autonomously, which could lead to further stress and burden on them [1]. On the other hand, physicians emphasize the need for supportive networks, a mechanism to deal with errors after obtaining assistance, and how clear disclosure instructions should be laid down. In contrast, nurses have expressed concerns about ambiguity in organizational cultures, as well as potential liability issues that add to their fear of possible retribution [11]. Undoubtedly, HCPs, including nurses, frequently face numerous challenges that prevent them from being open about their mistakes, as confirmed by research findings [12].

### 1.2. Significance of the Research

There is evidence that if patient safety incidents are disclosed, they will avoid unnecessary legal costs, costs of litigation, and, in some cases, negligence findings against practitioners. However, there are no studies that have specifically focused on the DPSI for the nursing profession, despite the fact that nurses are in frequent contact with medical errors and issues related to patient safety. Barriers to the open and honest disclosure of incidents should be identified, such that honesty in the face of error can be encouraged. This research may inform training programs, policy changes, and support systems to create a culture of safety. Thus, the ongoing study takes on a problem of paramount consideration in the domain of patient safety, healthcare quality, and ethical considerations.

This study highlights the barriers that discourage nurses from divulging medical errors and supports the Vision 2030 strategy in the Kingdom of Saudi Arabia in developing public service sectors, including health, and enhancing life [13]. It is concerned with patient safety, instilling confidence in providers and patients, and maintaining exemplary standards of care. The research also fosters a continuous improvement in culture and ethical practice in the health sector, which is critical for attaining the qualitative healthcare services envisioned in Vision 2030’s ambitious goals.

### 1.3. Research Questions

i.What are the barriers to error disclosure among nurses working at a tertiary hospital in the Kingdom of Saudi Arabia?ii.How do these barriers relate to the sociodemographic characteristics of the nurses working at a tertiary hospital in the Kingdom of Saudi Arabia?

## 2. Materials and Methods

### 2.1. Study Design and Setting

The researchers used a cross-sectional design. This research was carried out at King Fahad Hospital (KFH), Hofuf, which is the main hospital of Al-Ahsa Region in Eastern Saudi Arabia. KFH was accorded an accreditation from the Ministry of Health as a reference facility in 2020 for responding to nuclear and radiological accidents. It opened in the year 1400 Hijri (1980 AD) with a bed capacity of 502, acting as a secondary general hospital to 62 primary care health centers and 6 other regional hospitals. KFH was chosen because it has a high bed capacity relative to its nursing staff. It is noteworthy to highlight that nurses in Saudi Arabia are exposed to patient safety training in both their years of study, and during their new staff orientation, the nurse-to-patient ratio differs from one department to another. The critical care units (ICUs), on the one hand, have a one-to-one nurse-to-patient ratio, while, in other general wards, the ratio may reach up to one to eight. KFH serves to provide a wide spectrum of major specialties, several subspecialties, and specialized centers, such as the Prince Sultan Hospital for Cardiac Surgery. In addition, a new building for emergency care has recently been established. Additionally, KFH also received the Joint Commission International (JCI) accreditation in 2014 and the Central Board for Accreditation of Healthcare Institutions (CBAHIs) accreditation in 2017.

### 2.2. Sampling

The inclusion criteria included registered nurses (of different ages, educational levels, and from both sexes) who had provided direct patient care for at least one year of work and had agreed to participate in this study. The exclusion criteria included nurses who were not involved in the direct care of patients (such as head nurses, nurse managers, and supervisors) and who were unable to complete this study.

Convenience (nonprobability) sampling was used to obtain the study sample. Convenience sampling was chosen due to the descriptive nature of this study. A convenience sample of 255 nurses was drawn from a targeted population of 750 nurses. This sample size was determined using the Raosoft power analysis program (http://www.raosoft.com/samplesize.html, accessed on 1 January 2025), with a margin of error of 5% and a response distribution of 50%, at a confidence level of 95%.

### 2.3. Ethical Considerations

This study received ethical approval from the Research and Ethics Committee of the Faculty of Nursing at King Abdul-Aziz University, Jeddah, Ref No. 2M. 17. This study was, therefore, approved by the research ethics committee of the second health cluster in the Al-Ahsa Region for implementation in MOH hospitals, IRB-KFHH No. (H-05-HS-065). An official administrative approval was granted for conducting the research through KFH management. The privacy, anonymity, and confidentiality of the data were maintained and assured by obtaining the participants’ agreement to participate in the research before data collection. All of the participants’ information was anonymized and stored, and the encrypted database was only accessible to authorized researchers.

### 2.4. Study Tools

A structured questionnaire was used to collect the data. The questionnaire consisted of two parts, as detailed below.

#### 2.4.1. Sociodemographic Characteristics

Questions on sociodemographic characteristics (age, gender, marital status, level of education, and nationality) were included. Another set of questions was related to the nurses’ clinical experience (including unit assignments and years of experience).

#### 2.4.2. The Barriers to Error Disclosure Assessment (BEDA) Tool

The Barriers to Error Disclosure Assessment (BEDA) Tool, which was developed by Welsh et al. (2018), exhibits potential as a tool for identifying and measuring the disclosure barriers that might prohibit transparency and honesty in the therapeutic relationship between a provider and a patient [14].

The tool consisted of 30 items. The first 4 questions collected demographic information regarding the respondents’ training needs and experience with medical error disclosure. These 4 questions were answered with “yes” or “no”; under the second question, there were 5 sub-items, which were also answered with “yes” or “no”. There were 26 items describing specific barriers that inhibit disclosure by health care providers, which were further divided into 5 types of barriers: Barrier One—confidence and knowledge barriers (5 items); Barrier Two—institutional barriers (4 items); Barrier Three—psychological barriers (9 items); Barrier Four—financial concern barriers (3 items); and Barrier Five—other barriers (5 items). In the first part, each item was assigned a code of 1 for “yes” and 0 for “no”. In the second part, participants rated their level of agreement with each item using a 5-point Likert scale, ranging from 1 (strongly disagree) to 5 (strongly agree). Responses to the survey were classified into three distinct categories based on a statistical analysis of the mean scores, where a higher score indicates a greater barrier to error disclosure: low barrier (scores below 62), average barrier (scores ranging from 62 to 93), and high barrier (scores exceeding 93).

Cronbach’s alpha was used to assess the reliability of the survey items, with an overall alpha of 0.89 for the study. The researchers conducted a pilot study testing for internal consistency within the barrier questionnaire on 26 nurses.

### 2.5. Data Collection

Data collection took place between June 2024 and August 2024. The researchers collected the data using an online questionnaire. The process of data collection in the present study was performed by the researchers, who prepared an electronic questionnaire using Google Forms.

The study aim was defined and was accompanied by measures to ensure the confidentiality and anonymity of the participants—in other words, it was ensured that no identification of any kind could be made on any one of the participants during the study. Informed consent was obtained from every participant. Tools were offered in both Arabic and English so that non-Arabic-speaking nurses could be included. The principal researcher introduced the study to the nursing education department. Their head nurses sent the online questionnaire via WhatsApp application to the units’ groups.

### 2.6. Data Analysis

In this study, all of the data analyses were carried out using the Statistical Package for Social Sciences, version 26 (Armonk, NY, USA, IBM Corporation). Categorical variables were shown as numbers and percentages. Continuous variables were calculated and given as the mean and standard deviation. The association between the barrier to error disclosure score and the socio-demographic characteristics of the nurses was performed using the Mann–Whitney Z-test and the Kruskal–Wallis H-test. Normality tests were performed using the Shapiro–Wilk test and Kolmogorov–Smirnov test. Based on the results, the barrier to error disclosure score followed a non-normal distribution. Thus, non-parametric tests were applied. Furthermore, post hoc analysis was performed to determine the mean differences of the barrier to error disclosure score in relation to the unit of assignment. Statistical significance was set to *p* < 0.05.

## 3. Results

### The Participants’ Demographic Characteristics

Table 1 illustrates the frequency distributions and percentages of the sociodemographic characteristics of the nurses. The socio-demographic profiles of the nurses are represented in a tabular form. In total, 46.6% were aged between 21 and 30, and the majority were females (92.2%). The predominant unit to which the nurses were assigned was the surgical ward (23.1%), followed by the medical unit (19.2%).

According to Figure 1, when medical errors occur, only 18% of the nurses disclosed them to patients, the patient’s family, or to the patient’s significant other, while the majority (82%) did not. Additionally, 34.5% of the respondents reported having disclosed medical errors to another team of healthcare professionals, while the remaining 65.5% did not disclose errors (Figure 2). Barriers to the disclosure of errors and their association with sociodemographic characteristics of nurses will be highlighted next.

Through exploring the association between the barriers to disclosure score and the socio-demographic characteristics of the nurses, it was found that higher barrier scores were more likely to be associated with male nurses (Z = 2.506; *p* = 0.012), those working in an emergency or outpatient unit (H = 9.136; *p* = 0.028), and those who had not received any training in medical error disclosure (Z = 2.460; *p* = 0.014). One possible explanation for why nurses in the emergency department and outpatient care were less likely to disclose errors could be that they spend less time with patients compared to nurses in wards or intensive care units, who typically remain with patients for longer periods. No significant differences in barrier scores were observed in relation to age, nationality, educational level, and years of working experience in the nursing field (*p* > 0.05) (Table 2).

In Table 3, reverse-coded items were re-coded to align in accordance with the direction of the results, with the higher the score, the higher the barrier. This will eliminate bias in the calculation of the total score.

With respect to the assessment of barriers to error disclosure in Table 3, the statement that scored the highest and closer to 5 (strongly agree) in the confidence and knowledge barriers domain was “I am not sure how much I should disclose to a patient/family member in the event I am involved in a medical error” (3.31 ± 0.93), whilst the one with the lowest rating closer to 1 (strongly disagree) was “I am confident in my ability to disclose a medical error” (2.13 ± 0.86). For the domain of institutional barriers, the statement that was rated the highest was “My institution is supportive of the disclosure of medical errors by healthcare providers” (3.55 ± 0.95), while the statement with the lowest rating was “I receive mixed messages from my institution regarding the process of disclosing an error” (3.09 ± 0.97). Regarding the domain of psychological barriers, the top-rated statement was “Fear of losing patient trust” (3.64 ± 0.97), while “Fear of shame” had the lowest rating (3.24 ± 1.09). For financial concern barriers, the statement with the highest rating was “Fear of litigation” (3.28 ± 0.98), whereas “Fear of increased insurance premiums” had the lowest rating (3.17 ± 0.99). Finally, for the domain of other barriers, the statement with the highest rating was “I am afraid of being blamed for a medical error if I am not present during the disclosure of medical errors conversation with a patient and/or patient’s family” (3.47 ± 0.88), while the statement “I would like to be included in the error disclosure process in the event I was involved in a medical error” was found to have the lowest rating (2.42 ± 0.89).

Table 4 shows the descriptive statistics of the barriers to error disclosure. It was observed that the mean percentage score was higher for the domain of psychological barriers (68.2%), followed by the domain of institutional barriers (66.5%), and then the domain of financial concern barriers (64.5%), while the domain of confidence and knowledge barriers showed the lowest mean percentage score (55.2%). The total mean barrier score was 81.9 (SD 13.3), with a mean percentage score of 63%. Accordingly, the low, average, and high levels of barriers constituted 12.2%, 78.8%, and 9%, respectively (see also Figure 3).

## 4. Discussion

Error disclosure in nursing is crucial for patient safety and quality of care; however, different barriers prevent nurses from truly communicating errors to patients and families. In their study to explore the factors that facilitate or hinder error disclosure in Saudi tertiary hospitals, Alyaemni (2023) argued that although ethical, professional, and legal standards in Saudi Arabia manage and direct the disclosure of medical errors, factors including lack of knowledge and training, as well as personal beliefs, affect disclosure [15]. This study, with primary objectives, will analyze socio-demographic and clinical experience data with a basic focus on these barriers and their domains. This study found an association between barriers to error disclosure and socio-demographic factors among nurses, whereby male nurses scored higher on such barriers than female nurses. The under-representation of male nurses in the sample might also impact the study findings, as the trend of the nursing workforce continues to be female-dominated globally [16].

Cultural ascription may contribute to the rising score of barriers among male nurses. In their study, Zaghloul et al. (2016) found that men were more inclined to disclose errors in general, regardless of their history of any past error experience. This disparity is attributed to cultural, social, or psychological factors wherein male nurses view error disclosure as a threat to their integrity, while female nurses might be more free to disclose such errors because of their caring and empathetic nature. This also proves the need to investigate how systemic and contextual factors contribute to these gender differences in error disclosure [17].

Statistically, this study showed no significant difference in the barrier scores concerning age, nationality, education, or years of experience. This is contrary to the expectations that such relationships must exist across different healthcare contexts. Khachian et al. (2022) stated that contract nurses tend to feel less free than permanent nurses. There was no correlation found between the type of employment and perception of barriers in a recent study [5]. While prior studies have found that job title, gender, and history of errors affect disclosure behaviors, physicians are less inclined than non-physicians to disclose errors [18]. Furthermore, Zaghloul et al. (2016) stated that physicians are less likely to disclose errors compared to non-physicians, indicating that job title affects disclosure behaviors, as shown by the results. Such variations can be seen as grounded in underlying cultural influences on disclosure attitudes [17]. These discrepancies can be explained by various cultural influences on attitudes toward disclosing.

Additionally, nurses in emergency and outpatient units reported experiencing more impediments to error disclosure than nurses in other units. In their study, Giraldo et al. (2017) showed that high-stress environments were correlated with major medical errors. These environments encourage rapid decision making and have large patient flows that induce more stress and feelings of fear that one will be judged or punished [19].

This study showed that nurses face moderately high barriers to disclosing medical errors, with 63% of the respondents reporting barriers to medical error disclosure. Specifically, just 12.2% indicated low barriers, 78% reported moderate barriers, and 9% reported high barriers. On the other hand, in their findings, Khachian et al. (2022) showed that nurses perceived more facilitators for the disclosure of medical errors and considered negative repercussions less significant, highlighting some other aspects [5]. Those nurses might have viewed their work environment more positively compared to our participants, who indicated much greater challenges in this context.

This study analyzed barriers to error disclosure, concluding that psychological barriers rank highest, followed by institutional barriers and then financial concerns. Confidence and knowledge barriers scored the lowest. While most studies identified similar barriers, their rankings might vary based on the focus areas. These studies confirmed that the main barriers faced by healthcare providers (HCPs) include significant psychological barriers [2,19,20,21]. Other research suggested that different rankings for error disclosure are based on the occupational group, type of error, and health care setting [22,23].

As indicated by McLennan et al. (2016), hurdles to error disclosure arise from issues of intrapersonal and interpersonal communication. These authors point to problems associated with doctor–patient relationships, peer-to-peer interactions, and systemic factors, such as culture, policy, and legal pressures. Such barriers reduce openness and psychological safety, and they adversely impact the quality of error disclosure. A number of earlier studies pointed out similar barriers faced by individuals, confirming that consistent challenges across different contexts have been acknowledged in studies. Psychological, organizational, and financial reasons have all been implicated in non-disclosure of errors. The diversity across studies arises from differences in health systems, workplace cultures, and between populations. Nevertheless, this diversity points to specific solutions with respect to general and/or specific disclosure barriers [23].

To clarify the different barriers faced by nurses, it is essential to pinpoint the concerns based on their own assessments. Among psychological barriers, the highest mean was obtained for “fear of losing patient trust”, meaning nurses often resisted disclosing information because of this fear. Studies have found that, for nurses, maintaining trust is a barrier to disclosure, while, for physicians, transparency enhances trust [24]. In other words, nurses generally worry that openness places the patient–provider relationship, which is invaluable when it comes to the quality of care and patient satisfaction, at risk.

Nevertheless, the lowest mean score recorded in the present study within the psychological barrier dimension was “fear of shame”. However, despite this, it is still considered to be a significant barrier. This sentiment coincides with the study conducted by Iedema et al. (2011), where it was explained that HCPs most likely believe patients will blame them as individuals for mistakes instead of understanding the true complexity of medical errors [25]. Again, emotional factors like shame and fear of negative reactions might discourage HCPs from being straightforward in sensitive situations. Thus, a culture of open dialogue and education on the importance of disclosure will enhance trust in the healthcare framework. Moreover, this notion parallels the findings of Carmichael T. (2017), which found that medical professionals hesitated to disclose errors due to fear of repercussions from colleagues. Such fears have built a culture in which the risks of disclosure outweigh the benefits [26]. Similarly, some nurses do not report errors for fear of negative consequences on their reputation, credibility, and risk of isolation in the workplace, thus creating standing challenges for transparency in health care. Furthermore, these results correspond with the study by Albalawi et al. (2020), where it was highlighted that blame culture and ineffective communication in Arab nations, including Saudi Arabia, are prevalent issues contributing to a weak patient safety culture. These communication challenges were attributed to workforce diversity, language barriers, and cultural differences. Among nurses in Saudi Arabia, the fear of blame was identified as a significant obstacle to reporting incidents and medication errors [27].

The second dimension preventing error disclosure among nurses, according to a recent study, is institutional barriers. The item that had the highest degree of agreement was “My institution does support healthcare providers in disclosing medical errors”. Thus, moderate support for error disclosure did exist for the study participants. There is an increased emphasis in the literature that a supportive institutional culture, strong leadership, and properly functioning organizational structures are required for effective error disclosure practices [28].

Conversely, the statement, “My institution sends mixed messages about how to disclose an error”, received the lowest score, yet its mean still held relevance. This suggests some confusion in the institution’s communication regarding error disclosure policies. According to Choi J et al. (2019) and Pyo J et al. (2020), the failure to report errors was attributed to high workloads and a lack of adequate resources [1,29]. HCPs are under massive conflicting pressures imposed by leaders regarding disclosure and handling sensitive situations [30]. Moreover, Walton M et al. (2019) noted that patients frequently revealed their experiences informally, thus making formal disclosure processes less viable in health institutions [31]. Conversely, Waller R et al. (2020) argued that nurses are encumbered with disclosure barriers, such as a lack of institutional support and the burden of disclosure errors [32].

After the above-mentioned two dimensions, an explicit barrier to error disclosure was found to be related to financial concerns. Another finding showed that “fear of litigation” obtained the highest rating and was deemed by the participants to be the most probable challenge to error disclosure among nurses; in contrast, the mean value for “fear of higher premiums” was the lowest, pointing to it as being somewhat less of an immediate concern.

Additional supporting evidence indicates that fear of lawsuits and the practices of insurers impede clinicians from disclosing errors [25]. Other studies further showed that HCPs are expected to maintain transparency but generally avoid disclosing errors due to their fear of malpractice lawsuits [3,22,33,34]. Additionally, Aubin L et al. (2022) further mentioned that surgeons are not willing to report even minor adverse events due to such fears [35]. Moreover, Alshaban, H et al. stressed that over half of nurses perceived error disclosure as a source from which lawsuits may derive [36].

In addition, Albalawi A et al. (2020) reported that the issue of blame culture is further underscored by the increasing number of complaints and legal claims filed against healthcare providers, often stemming from adverse events linked to patient mortality or morbidity. In Saudi Arabia, the Ministry of Health revealed that, between 2012 and 2015, 91% of legal claim cases were deemed preventable. This highlights the critical need for fostering a non-punitive environment to promote error reporting and improve patient outcomes [27].

In contrast, because of the possible worry of law that may complicate compromises, settlement agreements frequently encompass restrictive clauses of non-disclosure, with more than three-quarters of settlements prohibiting any acknowledgment of their existence. This trend, under the influence of tort reform, exemplifies a legal culture that is resistant to disclosure. In addition, Yan S et al. (2017) indicated that patients are concerned about complaint procedures and financial transparency in hospitals, which negatively affect physicians’ willingness to disclose errors due to fear of complaints [10].

Although new initiatives have been implemented, economic and legal problems still remain relevant today. Based on a study of Pennsylvania’s 2002 Act 13, the institution of this law was meant to improve error disclosures within healthcare organizations. It reported that disclosed malpractice claims involve much higher settlements than undisclosed ones, with the maximum involving claims based on severe injuries. The number of claims per 1000 admissions has not changed, which indicates continuing economic problems [37].

Thus, several programs have been implemented to target economic and legal problems in healthcare. Programs such as Communication, Apology, and Resolution (CARe) and the Collaborative Communication Resolution Program (CCRP) have demonstrated success in a marked reduction in malpractice claims and in cutting down resolution time. Painter R et al. (2018) estimated claims to have occurred in only 5% of incidents [38], while Mello M et al. (2017) saw a decrease in the time resolution from 17 to 8 months [39].

Thus, fear of lawsuits and financial repercussions discourage HCPs from communicating mistakes. While programs like CARe and CCRP promote open communication, the specter of legal consequences and financial loss persists. Reforms must go beyond the law to enhance protections for healthcare workers, improve insurance practices, and create a culture in which transparency is supported rather than punished.

Examining the barriers related to error disclosure, one major finding was that, according to slightly over half of the respondents, barriers relating to confidence and knowledge had the lowest score. The statement “I am not sure how much I should disclose to a patient/family member in the event I am involved in a medical error” showed a very high mean score. These findings are in agreement with a study by Iedema et al. (2011) wherein similar issues were raised, whereby providers frequently encountered uncertainty in making decisions regarding what details are to be shared after clinical errors have taken place [25].

The interrelation with information sharing has been highlighted by various studies. In patient participation, according to LeCraw J et al. (2018), the balancing act of patient safety on the one side and the legal protections and liability in assessing informed confidentiality on the other side makes it difficult to implement policies [40]. In addition, the disposition to withhold some information has been reported in other studies where there is an inclination for serious medical errors to be reported, while near misses are frequently not reported due to uncertainties as far as disclosure standards are concerned. It is commonly the severity of the error that usually determines whether it will be disclosed or not; with different cases, however, there is usually a broad variation among them [19,29,33,41].

On the other hand, there are some studies in support of disclosure. For instance, Mira J, et al. (2017) found that hospital pharmacists recommend full disclosure [42], while Carmichael T. (2017) noted that many doctors seek to disclose harmful errors. Still, many nurses are not sure of what to disclose, although many believe that it is important to disclose any potentially harmful errors, regardless of whether harm has occurred or not [26].

It has been noted that many studies have pointed out significant differences between how transparency is perceived in healthcare. Yan S et al. (2017) illustrated the gap between the expectations of healthcare providers and their consumers regarding transparency [10]. Moreover, Jones T et al. (2019) observed a conflict between the desire for transparency and a need for protection during disclosure [30]. In addition, Choi J et al. (2019) showed that Korean nurses recognize the importance of disclosure policies; however, they need enhanced training and clearer guidelines on disclosure [1], while Kim S et al. (2017) added the role played by the media in publicizing medical errors, adding difficulty to the process of disclosure [43].

Earlier studies have shown that, on the one hand, many acknowledge that error disclosure is an important art, while, on the other hand, the ways of performing it vary widely [24,30,32]. Such variation is influenced by several factors, such as the nature of the legal environment, cultural attitudes, and organizational reporting policies [3,6]. General agreement about disclosure practices, training for those working in the health field, and policies that guarantee some level of transparency, while also protecting the parties involved with the errors, are needed. Enhanced policies can ensure that disclosures are ethical and meet patients’ expectations for open communication [11].

Moreover, a lack of confidence is attributed to medical error disclosure in this study. Iedema R et al. (2011) revealed the challenge clinicians face in disclosing errors, particularly when confronted with an upset patient and their peers, as it requires high levels of poise to deal with this matter. In addition, Choi J et al. (2019) argued that there are no guidelines to govern the process of disclosing patient safety incidents. For many clinicians, the fear of disclosing errors often stems from emotional pressure from unhappy patients and worry about what might happen. Hence, a framework for improved communication among doctors regarding error disclosure needs to be developed [1]. Nonetheless, unlike our study, Musharrafieh U et al. (2019) established that unawareness and lack of confidence cause significant barriers for nurses who reported making errors (even though they did not necessarily disclose information about their errors because of these barriers), and this is coupled with a lack of support [44]. Different results were found in this study, perhaps due to variations in institutional support or training, and in the target groups recruited in previous studies compared to those in ours.

The last dimension that the study dealt with was termed “other barriers”. The top-ranked factor was fear of being blamed for making a medical error by not attending the meeting for disclosure of cases to patients or their families. In their findings, Aubin L et al. (2022) and Alshaban H et al. (2019) found that medical professionals often deflect disclosure of information because of the culture of defensive practice that fosters blame, shame, and fear of reprisal. In more explicitly drawn terms, nurses often find themselves under coercive, both emotional and institutional, pressures to refrain from disclosing errors, fearing that they may be blamed for the errors, especially when they are not in a privileged position in the decision-making process. To promote a less blameful culture and, therefore, better communication, institutional support needs to be strengthened or encouraged [35,36].

In addition, the lowest mean score related to “other barriers” was obtained for the belief that “the physician is ultimately responsible for disclosing the medical error, regardless of his or her involvement”. Thus, responsibilities for disclosing errors in healthcare settings require further exploration. Perez M et al. (2014) stated that it is the organization and not the individual nurse who is accountable for the error that has occurred [24]. Furthermore, according to Mira J et al. (2017), some HCPs do not agree on who should take the lead in disclosing patient safety incidents [42]. Additionally, Choi J et al. (2017) noted that certified nurses acknowledged the importance of disclosure, yet were not in consensus regarding the primary responsibility [1]. In addition, Mohamed M et al. (2023) reported that nurses had higher awareness of what errors were and to whom they should report those errors than HCPs [45]. Moreover, Walton M et al. (2019) reported that most informal disclosures were made by nurses and that medical practitioners were responsible for fewer than a quarter of informal disclosures [31]. On the other hand, Gu Q. and Deng Y. (2021) argued that expectation regarding disclosure varies according to the level of harm: Moderate harm should be disclosed by HCPs, whereas, for serious harm, top management must be involved. This shows that the nurses have ambiguous definitions of their roles in the disclosure of medical errors, which, in turn, place conflicting practices on frontline healthcare providers [46]. Organizations must, thus, support the roles of nurses and clarify responsibilities for error disclosure to avoid ambiguity.

Overall, this study extends current knowledge around barriers to medical error disclosure that are attributable to individual characteristics, such as the profession and nature of the working environment, as well as organizational factors, such as safety culture. This study shows that the practices of revealing errors vary significantly based on specific circumstances. Nurses in an emergency setting, as opposed to an outpatient setting, are less likely to disclose errors due to the immediacy of the consequences or due to the lack of support afforded in such dynamic environments. Job security can also be an influencing factor on individuals’ willingness to disclose errors, as those in more secure jobs often feel a greater commitment to their organization.

## 5. Limitations of the Study

The limitations of this research must be acknowledged since several were out of the control of the researchers. First, the findings may not represent the whole situation of the country. Another limitation is related to data collection, which relied on a self-report questionnaire and might have influenced the respondents to rate themselves overly positively and provide socially acceptable responses instead of more honest ones. Additionally, another point worth mentioning is that this study was limited by the low level of participation of male nurses. Furthermore, the use of a cross-sectional design and a lack of qualitative data limit the study’s ability to establish causality. Therefore, a different approach is recommended to provide deeper insights into nurse experiences.

## 6. Strengths of the Study

This study constitutes vital research in the area of healthcare since it involves barriers that may impede nursing in reporting medical errors, thus influencing patient safety and the quality of care. This study focused on the perspective of nurses, a group that is often under-represented in this area. In fact, according to recent research, it is the first study conducted in Saudi Arabia aimed at assessing barriers to error disclosure among nurses. This study sought to collect high-quality data using validated research instruments such as validated surveys and to uphold ethical principles in promoting honesty in reporting. The results of this research may inform training programs, policy changes, and support systems that should be in place to create a culture of safety, as well as help with removing certain barriers, such as the fear of blame.

## 7. Conclusions

Overall, this study provides important insights into the socio-demographic characteristics of nurses and the barriers they may encounter in disclosing medical errors. Gender, unit assignment of the nurse, and level of training received in medical error disclosure were factors contributing to the level of incident disclosure to patients and their families. Next, when medical errors occur, only a small group of nurses have successfully disclosed them to either patients and their families or another team of healthcare professionals. Data suggested that psychological, financial concern, and institutional barriers were identified as relating to error disclosure.

Targeted interventions are required to address the identified specific needs to support healthcare providers, specifically nurses. New regulations and policy changes are crucial for training programs implementation, enhancing safety culture, and tackling job-related insecurities to minimize barriers to error disclosure and ultimately provide better patient care quality.

## Figures and Tables

**Figure 1 healthcare-13-00831-f001:**
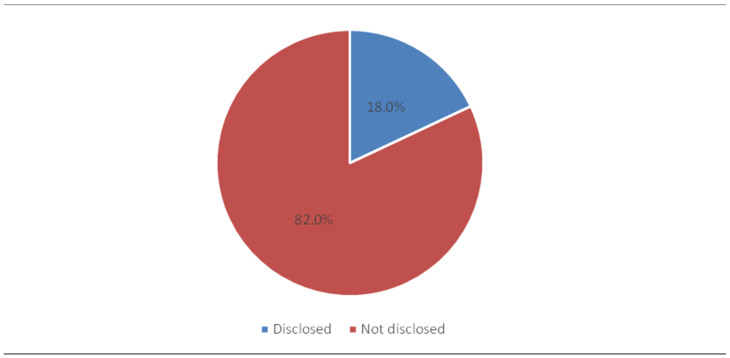
Have you ever disclosed a medical error to a patient, their family, or their significant other?

**Figure 2 healthcare-13-00831-f002:**
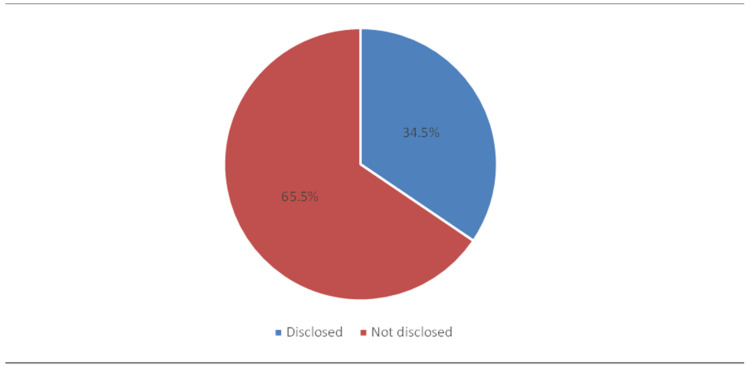
Have you ever disclosed a medical error with a team of other healthcare workers?

**Figure 3 healthcare-13-00831-f003:**
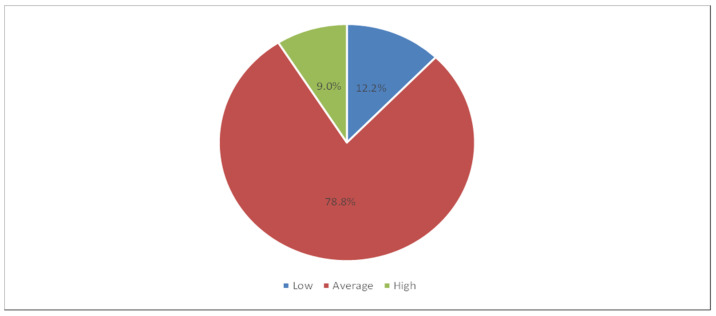
The levels of the barriers to error disclosure.

**Table 1 healthcare-13-00831-t001:** The socio-demographic characteristics of the nurses in this study.

Study Variables	*n* (%)
Age group	
21–30 years	119 (46.6%)
31–40 years	108 (42.4%)
41–50 years	27 (10.6%)
>50 years	1 (0.40%)
Gender	
Male	20 (7.8%)
Female	235 (92.2%)
Nationality	
Saudi	128 (50.2%)
Non-Saudi	127 (49.8%)
Educational level	
Diploma holder	38 (14.9%)
Bachelor’s degree	206 (80.8%)
Master’s degree	11 (4.3%)
Years of working experience in the nursing field	
1–5 years	116 (45.5%)
6–10 years	74 (29.0%)
11–15 years	35 (13.7%)
16–20 years	25 (9.8%)
>20 years	5 (2.0%)
Unit of assignment	
Surgical ward	59 (23.1%)
Medical units	49 (19.2%)
Emergency unit	41 (16.1%)
Critical care	37 (14.4%)
Outpatient department	23 (9.0%)
Isolation ward	13 (5.1%)
Hemodialysis unit	12 (4.7%)
Dental unit	06 (2.4%)
Oncology unit	06 (2.4%)
ICN/Nursery	03 (1.2%)
Operating room	03 (1.2%)
Others	03 (1.2%)
Total	255 (100%)

**Table 2 healthcare-13-00831-t002:** Association between barriers to disclosure, nurses’ socio-demographic characteristics, and previous levels of training received in medical error disclosure (*n* = 255).

Factor	Barrier Total Score (130) Mean ± SD	Z/H-Test	*p*-Value
Age group ^a^			
• ≤30 years	81.7 ± 13.4	0.064	0.949
• >30 years	82.0 ± 13.4		
Gender ^a^			
• Male	89.0 ± 11.9	2.506	**0.012 ****
• Female	81.3 ± 13.3		
Nationality ^a^			
• Saudi	82.9 ± 13.7	1.385	0.166
• Non-Saudi	80.7 ± 12.9		
Educational level ^a^			
• Diploma holder	83.3 ± 15.1	0.569	0.569
• Bachelor or master degree	81.6 ± 13.0		
Years of working experience in the nursing field ^a^			
• ≤10 years	81.7 ± 13.1	0.693	0.488
• >10 years	82.5 ± 14.2		
Unit of assignment ^b^			
• Critical Care	81.2 ± 11.3	9.136	**0.028 ****
• Emergency or outpatient unit	85.6 ± 13.3		
• Ward	81.2 ± 13.5		
• Others	74.8 ± 15.1		
Have you received any level of training in medical error disclosure? ^a^			
• Yes	79.4 ± 12.8	2.460	**0.014 ****
• No	84.0 ± 13.5		

^a^ *p*-value has been calculated using Mann Whitney Z-test. ^b^ *p*-value has been calculated using Kruskal Wallis H-test. ** Significant at *p* < 0.05 level.

**Table 3 healthcare-13-00831-t003:** Assessment of barriers to error disclosure (*n* = 255).

Confidence and Knowledge Barriers to Error Disclosure	Mean ± SD
1. I am comfortable with my ability to disclose a medical error *	2.17 ± 0.86
2. I am confident in my ability to disclose a medical error *	2.13 ± 0.86
3. I am not sure how much I should disclose to a patient/family member in the event I am involved in a medical error	3.31 ± 0.93
4. I am not sure when I should disclose an error	3.04 ± 1.04
5. I am unsure of my role in a conversation disclosing errors to the patient and/or family members	3.15 ± 0.99
**Institutional barriers to error disclosure**	
6. My institution supports an atmosphere of transparency in error disclosure	3.53 ± 0.94
7. My institution supports the disclosure of medical errors by healthcare providers	3.55 ± 0.95
8. I receive mixed messages from my institution regarding the process of disclosing an error	3.09 ± 0.97
9. I receive mixed messages from my institution regarding what types of errors should be disclosed	3.12 ± 0.98
**Psychological barriers to error disclosure**	
10. Fear of disciplinary action	3.56 ± 1.01
11. Fear of losing patient trust	3.64 ± 0.97
12. Fear of losing colleague support	3.40 ± 1.03
13. Fear of personal failure	3.42 ± 1.09
14. Fear of losing self-esteem	3.37 ± 1.12
15. Fear of damaged reputation	3.47 ± 1.05
16. Fear or judgment from colleagues	3.35 ± 1.05
17. Fear of shame	3.24 ± 1.09
18. Fear that peers will question my competence	3.30 ± 1.05
**Financial concern barriers to error disclosure**	
19. Fear of litigation	3.28 ± 0.98
20. Fear of losing malpractice insurance coverage	3.23 ± 1.04
21. Fear of increased insurance premiums	3.17 ± 0.99
**Other barriers to error disclosure**	
22. I am uncertain about how to report errors/mistakes	3.18 ± 0.97
23. My institution provides peer support services that help providers deal with the emotional consequences of error *	2.69 ± 0.94
24. I would like to be included in the error disclosure process in the event I was involved in a medical error *	2.42 ± 0.89
25. The physician is the ultimate one responsible for disclosing the medical error, regardless of his/her involvement in the error *	2.60 ± 0.92
26. I am afraid of being blamed for a medical error if I am not present during the disclosure of medical errors conversation with a patient and/or the patient’s family	3.47 ± 0.88

* Reverse-coded item. Responses ranged from strongly disagree (coded with 1) to strongly agree (coded with 5).

**Table 4 healthcare-13-00831-t004:** Descriptive statistics of the barriers to error disclosure scores and domains (*n* = 255).

Barrier to Error Disclosure Domain	Mean ± SD
Confidence and knowledge barriers score	13.8 ± 3.08
Institutional barriers score	13.3 ± 2.72
Psychological barriers score	30.7 ± 8.13
Financial concern barriers score	9.68 ± 2.73
Other barriers score	14.4 ± 1.81
Total barrier score	81.9 ± 13.3
**Level of barrier**	***n* (%)**
Low (score < 62%)	31 (12.2%)
Average (score 62–93%)	201 (78.8%)
High (score > 93%)	23 (9.0%)

## Data Availability

The data presented in this study are available on request from the corresponding author due to ethical and privacy reasons.

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
