# Peer review of "Breaking the Silence: Barriers to Error Disclosure Among Nurses in the Kingdom of Saudi Arabia—A Cross-Sectional Study"

_healthcare, 2025, doi:10.3390/healthcare13070831_

Round 1

Reviewer 1 Report

Comments and Suggestions for Authors

It is my pleasure to review the manuscript entitled "Breaking the Silence: Barriers to Error Disclosure among Nurses at Kingdom of Saudi Arabia." In this article, the authors conducted a study among nurses to understand the perception and factors contributing to barriers in error disclosure. Barriers to error disclosure in nursing is an interesting area of research in the context of patient safety and healthcare quality.

I have a few questions and comments for the authors' consideration.

The problem statement, the significance of the study, and the literature review were not comprehensive enough and need a critical presentation of it. Also, needed to explain the scientific background and rationale for the investigation being reported and state-specific objectives. Up to date the reference, and focus to most recent references.

My comments are as follows:

1-     The title, "Breaking the Silence: Barriers to Error Disclosure among Nurses at Kingdom of Saudi Arabia," effectively reflects the main focus of the paper. It clearly indicates the population (nurses) and the subject matter (error disclosure barriers), providing an accurate preview of the research content. However, it could be more precise by specifying the type of study or methodology used, such as "A Cross-Sectional Study," to give readers a clearer expectation.

2-     Abstract: The abstract offers a concise overview of the study, mentioning the background, aim, methods, and key findings. However, it lacks detailed information on the sample, specific methodology, and detailed recommendations. The abstract could be improved by elaborating on the methodology, specifying the sample size, data collection tools, and summarizing the main recommendations more clearly.

3-     Introduction: The introduction identifies the problem of underreporting medical errors and its significance in patient safety. While the research problem is clearly stated, the rationale behind choosing Saudi Arabia as the study setting is not well-articulated. The authors could enhance this section by providing more background on the cultural and institutional factors in Saudi Arabia that might influence error disclosure.

The purpose of the study is clearly stated: to assess barriers to error disclosure among nurses. However, the introduction lacks a strong theoretical framework. The authors should explicitly mention any theoretical models or frameworks guiding their research, which would provide a stronger foundation for the study. Moreover, the literature review is somewhat outdated and lacks critical analysis. Incorporating more recent studies and discussing gaps in the existing literature would strengthen this section.

4-     Methods

Setting: The study was conducted at King Fahad Hospital in Hofuf, Saudi Arabia, a tertiary healthcare facility with diverse clinical settings. While this setting is appropriate, the authors do not discuss how the hospital's unique characteristics might influence the generalizability of the findings. Including a comparison with other hospitals or healthcare settings in Saudi Arabia would provide a more comprehensive understanding.

Sample: the authors do not provide sufficient details on the inclusion and exclusion criteria, which are critical for evaluating the sample's representativeness.

Study instrument: Data were collected using a structured questionnaire, including the Barriers to Error Disclosure Assessment (BEDA) Tool. While the tool's reliability and validity are referenced from previous studies, specific reliability metrics (e.g., Cronbach's alpha) are not provided. The authors should have conducted a pilot study to validate the tool within the Saudi context, as cultural differences may affect the tool's applicability.

Ethical considerations: the authors should elaborate on how they ensured data security and addressed potential ethical dilemmas related to the sensitive nature of error disclosure.

Data collection procedure: I suggest explaining data collection steps.

5- Results: the presentation of results lacks depth. The authors should have provided more detailed statistical analyses, such as regression models, to control for potential confounding variables.

6- Discussion: The discussion contextualizes findings within existing literature, highlighting agreements and discrepancies with previous studies. However, it lacks critical analysis. The authors tend to summarize findings rather than engage in a deeper discussion of their implications. They should explore why certain barriers are more prominent in the Saudi context and consider alternative explanations for unexpected results.

Moreover, the discussion does not adequately address the study's limitations. For example, the potential impact of social desirability bias on self-reported data is not considered. The authors should also discuss the implications of their findings for different stakeholders, such as policymakers, healthcare administrators, and educators.

7- Conclusion: Add critical findings and conclusion.

8- Implications: The study discusses implications for nursing practice, emphasizing the need for training and institutional support to reduce disclosure barriers. However, these recommendations are somewhat generic. The authors should provide specific, actionable strategies for implementing their recommendations, such as targeted training programs or policy changes.

9- Limitation: The authors acknowledge limitations, including potential response bias and limited generalizability. However, this section is superficial. They should discuss additional limitations, such as the cross-sectional design's inability to establish causality and the lack of qualitative data to provide deeper insights into nurses' experiences.

Comments on the Quality of English Language

The quality of the English language in the manuscript is generally clear and comprehensible. However, several areas could benefit from improvements to enhance clarity, coherence, and professionalism:

  1. Grammar and Syntax: There are occasional grammatical errors and awkward sentence structures that may confuse readers. Simplifying complex sentences and ensuring subject-verb agreement would improve readability.

  2. Consistency: Inconsistent terminology (e.g., "barriers to error disclosure" vs. "error disclosure barriers") should be standardized throughout the manuscript.

  3. Vocabulary: Some sections could benefit from more precise word choices to convey ideas more effectively. Replacing vague terms with specific academic vocabulary would strengthen the manuscript.

  4. Flow and Cohesion: Transitions between paragraphs and sections could be smoother. Adding linking phrases would improve the logical flow of ideas.

  5. Punctuation: Minor punctuation errors, such as missing commas and incorrect use of semicolons, should be corrected to enhance readability.

Author Response

Comment 1: The title, "Breaking the Silence: Barriers to Error Disclosure among Nurses at Kingdom of Saudi Arabia," effectively reflects the main focus of the paper. It clearly indicates the population (nurses) and the subject matter (error disclosure barriers), providing an accurate preview of the research content. However, it could be more precise by specifying the type of study or methodology used, such as "A Cross-Sectional Study," to give readers a clearer expectation.
Response 1:“A Cross-Sectional Study” was added to the title

Comment 2: Abstract: The abstract offers a concise overview of the study, mentioning the background, aim, methods, and key findings. However, it lacks detailed information on the sample, specific methodology, and detailed recommendations. The abstract could be improved by elaborating on the methodology, specifying the sample size, data collection tools, and summarizing the main recommendations more clearly.
Response 2: Sample, sample size and tool used were identified in (lines 23-25) and Recommendations were added to the abstract (lines 31-33)

Comment 3:

Introduction: The introduction identifies the problem of underreporting medical errors and its significance in patient safety. While the research problem is clearly stated, the rationale behind choosing Saudi Arabia as the study setting is not well-articulated. The authors could enhance this section by providing more background on the cultural and institutional factors in Saudi Arabia that might influence error disclosure.

The purpose of the study is clearly stated: to assess barriers to error disclosure among nurses. However, the introduction lacks a strong theoretical framework. The authors should explicitly mention any theoretical models or frameworks guiding their research, which would provide a stronger foundation for the study. Moreover, the literature review is somewhat outdated and lacks critical analysis. Incorporating more recent studies and discussing gaps in the existing literature would strengthen this section.
Response 3: 

  • The rationale behind choosing Saudi Arabia as the study setting was provided under the strength of the study (line 522)
  • Due to limited studied of the subject in general and in Saudi Arabia in particular, some references used were dated from 2012 up to recently to capture the different factors contributing to medical error disclosure.

Comment 4: 

 Methods

Setting: The study was conducted at King Fahad Hospital in Hofuf, Saudi Arabia, a tertiary healthcare facility with diverse clinical settings. While this setting is appropriate, the authors do not discuss how the hospital's unique characteristics might influence the generalizability of the findings. Including a comparison with other hospitals or healthcare settings in Saudi Arabia would provide a more comprehensive understanding.

Sample: the authors do not provide sufficient details on the inclusion and exclusion criteria, which are critical for evaluating the sample's representativeness.

Study instrument: Data were collected using a structured questionnaire, including the Barriers to Error Disclosure Assessment (BEDA) Tool. While the tool's reliability and validity are referenced from previous studies, specific reliability metrics (e.g., Cronbach's alpha) are not provided. The authors should have conducted a pilot study to validate the tool within the Saudi context, as cultural differences may affect the tool's applicability.

Ethical considerations: the authors should elaborate on how they ensured data security and addressed potential ethical dilemmas related to the sensitive nature of error disclosure.

Data collection procedure: I suggest explaining data collection steps.
Response 4: 

  • King Fahad Hospital in Hofuf, Saudi Arabia is one of few general tertiary healthcare facilities that is known for its diverse clinical settings. Therefore, selecting it as a setting can draw a baseline for further research in other locations.
  • Inclusion and exclusion criteria were specified in the Materials and Methods (line 170)
  • Reliability and validity test results were highlighted (line 215)
  • Data collection steps were mentioned (line 221)
  • Ethical considerations were detailed (line 181)

Comment 5: Results: the presentation of results lacks depth. The authors should have provided more detailed statistical analyses, such as regression models, to control for potential confounding variables.
Response 5: 

  • A statistical service was used to analyze data according to study aims to provide answers to research questions including descriptive analysis presenting the participants' demographic characteristics and work experience and addressing the research questions.
  • The association between the barrier to error disclosure score and the socio-demographic characteristics of the nurses has been performed using the Mann-Whitney Z-test and the Kruskal Wallis H-test. Normality tests were performed using the Shapiro-Wilk test and Kolmogorov-Smirnov test. Based on the results, the barrier to error disclosure score follows the non-normal distribution. Thus, the non-parametric tests were applied. Further, post-hoc analysis was performed to determine the mean differences of barrier to error disclosure score in relation to unit of assignment. Statistical significance was set to p<0.05. All data analyses were carried out using the statistical package for social sciences, version 26

Comment 6: Discussion: The discussion contextualizes findings within existing literature, highlighting agreements and discrepancies with previous studies. However, it lacks critical analysis. The authors tend to summarize findings rather than engage in a deeper discussion of their implications. They should explore why certain barriers are more prominent in the Saudi context and consider alternative explanations for unexpected results.
Moreover, the discussion does not adequately address the study's limitations. For example, the potential impact of social desirability bias on self-reported data is not considered. The authors should also discuss the implications of their findings for different stakeholders, such as policymakers, healthcare administrators, and educators.
Response 6: 

  • Further elements were added to highlight the significance of the Saudi culture on the matter (line 378, 439)
  • Different stallholders were mentioned in the implemention of recommendations were in the strength of the study (line 539)
  • Additional limitations were mentioned (line 529)
  • the potential impact of social desirability bias on self-reported data was mentioned in the limitation section (line 536)

Comment 7: Conclusion: Add critical findings and conclusion.
Response 7: Conclusion provided key findings and conclusion

Comment 8: Implications: The study discusses implications for nursing practice, emphasizing the need for training and institutional support to reduce disclosure barriers. However, these recommendations are somewhat generic. The authors should provide specific, actionable strategies for implementing their recommendations, such as targeted training programs or policy changes.
Response 8: implementing their recommendations were highlighted in the strength of the study (line 539)

Comment 9: Limitation: The authors acknowledge limitations, including potential response bias and limited generalizability. However, this section is superficial. They should discuss additional limitations, such as the cross-sectional design's inability to establish causality and the lack of qualitative data to provide deeper insights into nurses' experiences.
Response 9: Limitation section was modified (line 554)

Comment 10: 

Comments on the Quality of English Language
The quality of the English language in the manuscript is generally clear and comprehensible. However, several areas could benefit from improvements to enhance clarity, coherence, and professionalism:

  1. Grammar and Syntax: There are occasional grammatical errors and awkward sentence structures that may confuse readers. Simplifying complex sentences and ensuring subject-verb agreement would improve readability.
  2. Consistency: Inconsistent terminology (e.g., "barriers to error disclosure" vs. "error disclosure barriers") should be standardized throughout the manuscript.
  3. Vocabulary: Some sections could benefit from more precise word choices to convey ideas more effectively. Replacing vague terms with specific academic vocabulary would strengthen the manuscript.
  4. Flow and Cohesion: Transitions between paragraphs and sections could be smoother. Adding linking phrases would improve the logical flow of ideas.

Punctuation: Minor punctuation errors, such as missing commas and incorrect use of 
Response 10: Document underwent the journal’s professional English editing services

Reviewer 2 Report

Comments and Suggestions for Authors

The manuscript titled "Breaking the Silence: Barriers to Error Disclosure among Nurses in the Kingdom of Saudi Arabia" addresses a critical issue in healthcare: the challenges that nurses encounter when disclosing medical errors. The study offers valuable insights into the psychological, institutional, financial, and knowledge-based barriers that hinder transparency in error reporting.

Methodologically, the research is well-structured, utilizing a descriptive cross-sectional survey to evaluate the barriers to error disclosure among nurses at King Fahad Hospital in Hofuf. The employment of the Barriers to Error Disclosure Assessment (BEDA) Tool enhances the study's reliability by offering a structured framework for assessing the factors that influence disclosure behaviors. The sample size of 255 nurses is sufficient for statistical analysis, and the study effectively identifies significant sociodemographic trends that affect nurses' willingness to report errors.

The study offers practical contributions by providing specific recommendations for enhancing training, adjusting policies, and supporting institutions. Therefore, the findings are applicable for healthcare organizations.

Although the manuscript is insightful and well-executed, I have a few suggestions for improvement:

  1. While the study cites relevant literature, it would benefit from incorporating a specific theoretical model to contextualize the findings. Models such as Reason’s Swiss Cheese Model of Error Prevention or the Theory of Planned Behavior could offer a structured perspective for interpreting the barriers to error disclosure and their impact on nursing practice.
  2. The definition of “barriers” to error disclosure should be explicitly stated early in the introduction. Moreover, the discussion section mentions “intrapersonal and interpersonal communication barriers,” but these concepts are not clearly defined or explored in depth.
  3. It is noteworthy that male nurses report more significant barriers to disclosure than female nurses; however, the discussion does not fully examine the potential cultural or psychological reasons behind this trend. While existing studies on gender differences in healthcare are referenced, additional discussion on socio-cultural expectations in Saudi Arabia would offer deeper insights into this phenomenon.
  4. The study rightly acknowledges the ethical implications of error disclosure but lacks a thorough discussion of Saudi Arabia's current hospital policies regarding mandatory disclosure. Comparing these policies with international frameworks would strengthen the discussion and provide valuable context for evaluating Saudi Arabia’s regulatory landscape.

Author Response

Comment 1: While the study cites relevant literature, it would benefit from incorporating a specific theoretical model to contextualize the findings. Models such as Reason’s Swiss Cheese Model of Error Prevention or the Theory of Planned Behavior could offer a structured perspective for interpreting the barriers to error disclosure and their impact on nursing practice.
Response 1: 

Results were interpreted as follows: analysis of socio-demographic and clinical experience data, evaluation of training in medical error disclosure, identification of helpful aspects for learning about error disclosure, assessment of practices in error disclosure to patients, their families, or significant others, exploration of disclosing medical errors within a healthcare team, and examination of barriers to error disclosure and its domains.

Theoretical model to contextualize the findings such as Reason’s Swiss Cheese Model of Error Prevention or the Theory of Planned Behavior will be highly considered in our next paper

Comment 2: The definition of “barriers” to error disclosure should be explicitly stated early in the introduction. Moreover, the discussion section mentions “intrapersonal and interpersonal communication barriers,” but these concepts are not clearly defined or explored in depth.
Response 2: In the introduction section, the different types of barriers were highlighted including communicational and institutional ones with examples.

Comment 3: It is noteworthy that male nurses report more significant barriers to disclosure than female nurses; however, the discussion does not fully examine the potential cultural or psychological reasons behind this trend. While existing studies on gender differences in healthcare are referenced, additional discussion on socio-cultural expectations in Saudi Arabia would offer deeper insights into this phenomenon.
Response 3: Further elements were added to highlight the significance of the Saudi culture on the matter (lines 378, 413)

Comment 4: The study rightly acknowledges the ethical implications of error disclosure but lacks a thorough discussion of Saudi Arabia's current hospital policies regarding mandatory disclosure. Comparing these policies with international frameworks would strengthen the discussion and provide valuable context for evaluating Saudi Arabia’s regulatory landscape.
Response 4: Saudi Arabia's current hospital policies regarding mandatory disclosure was highlighted in the discussion (line 306)

Reviewer 3 Report

Comments and Suggestions for Authors

This paper examines barriers to disclosure of medical errors to patients amongst nurses at a Saudia Arabian hospital.  Although this is an important issue potentially impacting patient safety, there are a number of serious issues with the manuscript that must be addressed.

Abstract – the results and conclusions don’t really make sense and don’t adequately describe the findings of the study.  This must be rewritten.

Introduction – Clarification is needed regarding error disclosure.  Is this only to patients or to other members of staff as well?  Should the word “Patients” in the middle of line 96 be “nurses”?  The aim stated in line 104 doesn’t match the study.  Much of this section is also more appropriate in the discussion section.

Methods – abbreviations have been used without any explanation as to what they mean.  Much more detail and justification are needed in the study design and sampling sections.

Results – please reformat tables to left-justified.  Most of the results aren’t presented in a way that matches the suggested aim or conclusions.  Table 3 has statistical calculations on variables that don’t match those in table 2.  The “mean percentage score” doesn’t appear to correlate with any information given in any table – what does this mean?  Adding scores from domains with different numbers of questions will always result in different totals – why hasn’t a “domain average” been calculated to allow for comparison?  Statistical significance could then also be calculated to look for differences between groups (ranges should also be included).  Under table 3.3 is the note “reverse coded item”.  What does this mean?  How does this then impact your calculations?

Discussion – much of the discussion does not agree or relate to the results.  What does “just below one-fifth indicated low barriers” (line 317) mean?  Is this those who agreed or strongly agreed with the statements?  How does this relate to the “reverse coded items”?  Many statements throughout the discussion do not have support from the results.  This section requires extensive rewriting.

Conclusion – this doesn’t match the aims or results and needs to be rewritten (particularly after the results are redone).  Statements are being made about the socio-demographic characteristics and yet none of this has been analysed in the results.

Citing throughout – this must be reviewed, as many citations are noted inappropriately throughout the text (e.g. “Smith et al (2022) stated…..” not “[11] stated…..”).  There are also many areas throughout the manuscript where references are required.  Reference list should be reviewed for formatting errors/missing information.

Comments on the Quality of English Language

English language – must be extensively reviewed prior to any further submission.  The title should be clarified: is it “…in the Kingdom of Saudia Arabia” or is “Kingdom of Saudi Arabia” a hospital?  There are numerous grammatical, phrasing, and word choice errors throughout making the paper difficult to read

Author Response

Comment 1: Abstract – the results and conclusions don’t really make sense and don’t adequately describe the findings of the study.  This must be rewritten.
Response 1: Results and conclusion were rewritten (line 27, 31)

Comment 2: Introduction – Clarification is needed regarding error disclosure.  Is this only to patients or to other members of staff as well?  Should the word “Patients” in the middle of line 96 be “nurses”?  The aim stated in line 104 doesn’t match the study.  Much of this section is also more appropriate in the discussion section.
Response 2: 

  • the word “Patients” in the middle of line 101 was modified to “nurses”
  • the final paragraph in the introduction was modified (line 109)

Comment 3: Methods – abbreviations have been used without any explanation as to what they mean.  Much more detail and justification are needed in the study design and sampling sections.
Response 3: 

  • Inclusion and exclusion criteria were specified in the Materials and Methods (line 170)
  • Reliability and validity test results were highlighted (line 215)
  • Data collection steps were mentioned (line 221)

Comment 4: Results – please reformat tables to left-justified.  Most of the results aren’t presented in a way that matches the suggested aim or conclusions.  Table 3 has statistical calculations on variables that don’t match those in table 2.  The “mean percentage score” doesn’t appear to correlate with any information given in any table – what does this mean?  Adding scores from domains with different numbers of questions will always result in different totals – why hasn’t a “domain average” been calculated to allow for comparison?  Statistical significance could then also be calculated to look for differences between groups (ranges should also be included).  Under table 3.3 is the note “reverse coded item”.  What does this mean?  How does this then impact your calculations?
Comment 4: 

  • A statistical service was used to analyze data according to study aims to provide answers to research questions including descriptive analysis presenting the participants' demographic characteristics and work experience and addressing the research questions.
  • The association between the barrier to error disclosure score and the socio-demographic characteristics of the nurses has been performed using the Mann-Whitney Z-test and the Kruskal Wallis H-test. Normality tests were performed using the Shapiro-Wilk test and Kolmogorov-Smirnov test. Based on the results, the barrier to error disclosure score follows the non-normal distribution. Thus, the non-parametric tests were applied. Further, post-hoc analysis was performed to determine the mean differences of barrier to error disclosure score in relation to unit of assignment. Statistical significance was set to p<0.05. All data analyses were carried out using the statistical package for social sciences, version 26

Comment 5: Discussion – much of the discussion does not agree or relate to the results.  What does “just below one-fifth indicated low barriers” (line 317) mean?  Is this those who agreed or strongly agreed with the statements?  How does this relate to the “reverse coded items”?  Many statements throughout the discussion do not have support from the results.  This section requires extensive rewriting.
Response 5: Discussion section was rewritten and edited. further elements were added

Comment 6: Conclusion – this doesn’t match the aims or results and needs to be rewritten (particularly after the results are redone).  Statements are being made about the socio-demographic characteristics and yet none of this has been analysed in the results.
Response 6: The conclusion provided important insights into the socio-demographic characteristics of nurses and the barriers they encounter in disclosing medical errors. As the aim of the study was to assess barriers to error disclosure among nurses at the Kingdom of Saudi Arabia, the conclusion statement is relevant.

Comment 7: Citing throughout – this must be reviewed, as many citations are noted inappropriately throughout the text (e.g. “Smith et al (2022) stated…..” not “[11] stated…..”).  There are also many areas throughout the manuscript where references are required.  Reference list should be reviewed for formatting errors/missing information.
Comment 7: In-text reference use a Vancouver style and list was reviewed 

Comment 8: 

Comments on the Quality of English Language

English language – must be extensively reviewed prior to any further submission.  The title should be clarified: is it “…in the Kingdom of Saudia Arabia” or is “Kingdom of Saudi Arabia” a hospital?  There are numerous grammatical, phrasing, and word choice errors throughout making the paper difficult to read

Response 8: Document underwent the journal’s professional English editing services

Reviewer 4 Report

Comments and Suggestions for Authors

The title is appropriate, concise, and accurately reflects the study’s objective. It highlights the central theme—barriers to error disclosure among nurses—while specifying the geographical context, which enhances the study's relevance.

The abstract provides a clear overview of the study, including its objective, methodology, key findings, and conclusion. The inclusion of barrier percentages strengthens the transparency of the findings. However, it could benefit from a sentence highlighting the practical implications of the study for patient safety.

The introduction is well-structured and provides a solid theoretical foundation. It explains the importance of error disclosure in nursing and its relationship with patient safety. Additionally, the literature review is up to date and demonstrates the necessity of this study. The research question is well defined and justifies the investigation.

The methodology is detailed and well-structured. The study employs a descriptive cross-sectional design, which is appropriate for identifying barriers to error disclosure. The use of a convenience sample of 255 nurses is a limitation, as it may reduce the generalizability of the results. The application of the "Barriers to Error Disclosure Assessment (BEDA) Tool" is appropriate and well described, but more details on its validation within the studied population would be beneficial. The statistical analysis is rigorous, using non-parametric tests due to data distribution, demonstrating methodological robustness.

The results are presented clearly and objectively. The study identifies psychological, institutional, and financial barriers as the main challenges to error disclosure among nurses. The inclusion of tables and statistical data improves result comprehension. However, additional graphical representations could further enhance the visualization of differences between groups.

The discussion interprets the findings in light of existing literature and reinforces the study's relevance. The emphasis on psychological barriers, such as fear of losing patient trust, is consistent with previous research. Comparing the findings with previous studies strengthens their validity. However, including concrete strategies to mitigate these barriers would add value to this section.

The conclusion adequately synthesizes the key findings and highlights the need for targeted interventions to improve error communication in nursing. Additionally, the recommendation for specific training for nurses working in emergency and outpatient units is relevant and based on the results obtained.

The authors acknowledge the study’s limitations, including its cross-sectional nature and the use of a convenience sample. However, they could further explore how these limitations impact the interpretation of the findings.

The references are current, relevant, and support the arguments presented throughout the article. The inclusion of recent studies strengthens the theoretical foundation of the work.

The tables are well-structured and present the data clearly. However, adding graphs could make the interpretation of findings more intuitive.

Overal Opinion: The article makes a significant contribution to the literature on patient safety by identifying barriers nurses face in error disclosure. It is well-founded, methodologically rigorous, and presents relevant practical implications.

Recomendation: Accepted with minor revisions. It is recommended to include graphical representations to facilitate data interpretation and expand the discussion on strategies to mitigate the identified barriers.

Author Response

Comment1: The abstract provides a clear overview of the study, including its objective, methodology, key findings, and conclusion. The inclusion of barrier percentages strengthens the transparency of the findings. However, it could benefit from a sentence highlighting the practical implications of the study for patient safety.
Response1: Recommendations were added to the abstract

Comment2: The results are presented clearly and objectively. The study identifies psychological, institutional, and financial barriers as the main challenges to error disclosure among nurses. The inclusion of tables and statistical data improves result comprehension. However, additional graphical representations could further enhance the visualization of differences between groups.
Response2: Additional graphs were added (line 258) 

Comment3: The discussion interprets the findings in light of existing literature and reinforces the study's relevance. The emphasis on psychological barriers, such as fear of losing patient trust, is consistent with previous research. Comparing the findings with previous studies strengthens their validity. However, including concrete strategies to mitigate these barriers would add value to this section.

Response3: Few additional items were added

Comment 4: The conclusion adequately synthesizes the key findings and highlights the need for targeted interventions to improve error communication in nursing. Additionally, the recommendation for specific training for nurses working in emergency and outpatient units is relevant and based on the results obtained.
Response 4: Conclusion section was modified to add further recommendations 

Comment 5: The authors acknowledge the study’s limitations, including its cross-sectional nature and the use of a convenience sample. However, they could further explore how these limitations impact the interpretation of the findings.
Response 5: Limitation section was modified (line 549)

Comment 6: The tables are well-structured and present the data clearly. However, adding graphs could make the interpretation of findings more intuitive.
Response 6: Additional graphs were added (line 258) 

Comment 7: Recomendation: Accepted with minor revisions. It is recommended to include graphical representations to facilitate data interpretation and expand the discussion on strategies to mitigate the identified barriers.
Response 7: All recommendations were considered, and modifications were made accordingly 

Round 2

Reviewer 3 Report

Comments and Suggestions for Authors

Many of the issues identified have not been addressed:

Introduction – much of this section is more appropriate in the discussion section – it is discussing conclusions not introducing the topic.

Methods – abbreviations have been used without any explanation as to what they mean.  More detail and justification are needed in the study design and sampling sections.  There is no mention of how “reverse coded items” in table 3 are being calculated/analysed.

Results – please reformat table 1 to left-justified.  Most of the results aren’t presented in a way that matches the suggested aim or conclusions.  Table 3 has statistical calculations on variables that don’t match those in other tables.  No further detail has been included for table 4.  The “mean percentage score” doesn’t appear to correlate with any information given in any table – what does this mean?  Adding scores from domains with different numbers of questions will always result in different totals – why hasn’t a “domain average” been calculated to allow for comparison?  Statistical significance could then also be calculated to look for differences between groups (ranges should also be included). 

Discussion – apart from improving the grammar, very little has been changed, and much of the discussion does not agree or relate to the results.  What does “just below one-fifth indicated low barriers” mean?  Is this related to table 4 - as this table states 12.2%?  How does this relate to the “reverse coded items”?  Many statements throughout the discussion do not have support from the results.  This section requires extensive rewriting.

Conclusion – this doesn’t match/are not supported by the results and needs to be rewritten.  

Citing throughout – this must be reviewed, as many citations are noted inappropriately throughout the text (e.g. “Smith et al (2022) stated…..” not “[11] stated…..”).  There are also many areas throughout the manuscript where references are required.  Reference list should also be reviewed for formatting errors/missing information.

Author Response

Comment 1: Introduction – much of this section is more appropriate in the discussion section – it is discussing
Responce 1: The authors attempted to highlight the dimensions of the matter through the available data in the literature. No results of the study were mentioned nor compared to in this section. 

Comment 2: conclusions not introducing the topic.
Responce 2: Conclusion was modified

Comment 3: Methods – abbreviations have been used without any explanation as to what they mean. More detail and justification are needed in the study design and sampling sections. There is no mention of how “reverse coded items” in table 3 are being calculated/analysed.
Responce 3: 

  • Abbreviations were explained
  • A detailed information about the setting was provided to justify the sample location choice.
  • Reverse coded items are necessary to be re-coded as the result should be according to the barriers, with the higher the score the higher the barrier, if we dont recode the negative items, this will not be in accordance to that direction. It should be going to positive direction to avoid bias in the calculation of total score

    Comment 4: Results – please reformat table 1 to left-justified. Most of the results aren’t presented in a way that matches the suggested aim or conclusions. Table 3 has statistical calculations on variables that don’t match those in other tables. No further detail has been included for table 4. The “mean percentage score” doesn’t appear to correlate with any information given in any table – what does this mean? Adding scores from domains with different numbers of questions will always result in different totals – why hasn’t a “domain average” been calculated to allow for comparison? Statistical significance could then also be calculated to look for differences between groups (ranges should also be included).
    Responce 4: 
  • Table format was modified
  • This study measures the overall barriers of the nurses, it didn’t ask about the specific barriers.
  • Mean and SD was used for easier interpretation to look which groups has the higher mean scores that’s why z-test has been used to determine the differences in scores.
  • Percentage scores was used  to compare which domains shows better than the others as comparing by mean would not be appropriated due to differences in the number of questions per domain. This representation doesn’t imply to correlate with other information, it just to show which domains is higher based on percentage scores. However, all measures are presented in table 3 (mean, sd, median, min, max) for domains and the overall score.

    Comment 5: Discussion – apart from improving the grammar, very little has been changed, and much of the discussion does not agree or relate to the results. What does “just below one-fifth indicated low barriers” mean? Is this related to table 4 - as this table states 12.2%? How does this relate to the “reverse coded items”? Many statements throughout the discussion do not have support from the results. This section requires extensive rewriting.
    Respond 5: 
  • Change statement to ( just 12.2% of respondents indicated low barriers, which is notably lower than one-fifth (20%)) . values have been changed to percentages throughout the paper
  • Reverse coded items are necessary to be re-coded as the result should be according to the barriers, with the higher the score the higher the barrier, if we dont recode the negative items, this will not be in accordance to that direction. It should be going to positive direction to avoid bias in the calculation of total score

    Comment 6: Conclusion – this doesn’t match/are not supported by the results and needs to be rewritten.
    Responce 6: Conclusion was modified

    Comment 7: Citing throughout – this must be reviewed, as many citations are noted inappropriately throughout the text (e.g. “Smith et al (2022) stated.....” not “[11] stated.....”). There are also many areas throughout the manuscript where references are required. Reference list should also be reviewed for formatting errors/missing information.
    Responce 7: Citation was modified when applicable
